

# Selection on $X_1 + X_2 + \cdots + X_m$ via Cartesian product trees

Patrick Kreitzberg[1], Kyle Lucke[2], Jake Pennington[1] and Oliver Serang[2]

[1] Department of Mathematics, University of Montana, Missoula, MT, United States of America
[2] Department of Computer Science, University of Montana, Missoula, MT, United States of America

## ABSTRACT

Selection on the Cartesian product is a classic problem in computer science. Recently, an optimal algorithm for selection on $A + B$, based on soft heaps, was introduced. By combining this approach with layer-ordered heaps (LOHs), an algorithm using a balanced binary tree of $A + B$ selections was proposed to perform selection on $X_1 + X_2 + \cdots + X_m$ in $o(n \cdot m + k \cdot m)$, where $X_i$ have length $n$. Here, that $o(n \cdot m + k \cdot m)$ algorithm is combined with a novel, optimal LOH-based algorithm for selection on $A + B$ (without a soft heap). Performance of algorithms for selection on $X_1 + X_2 + \cdots + X_m$ are compared empirically, demonstrating the benefit of the algorithm proposed here.

# INTRODUCTION

Sorting all values in $A + B$, where $A$ and $B$ are arrays of length $n$ and $A + B$ is the Cartesian product of these arrays under the $+$ operator, is nontrivial. In fact, there is no known approach faster than naively computing and sorting them which takes $O(n^2 \log(n^2)) = O(n^2 \log(n))$ time (*Bremner et al., 2006*); however, *Fredman (1976)* showed that $O(n^2)$ comparisons are sufficient, though no $O(n^2)$ time algorithm is currently known. *Frederickson & Johnson (1982)* published an algorithm which selects the $k^{th}$ value in $A + B$ with runtime $O(n + p \log(k/p))$ where $p = min(k, n)$. In 2018, Kaplan *et al.* described an optimal method for selecting the top $k$ values of $A + B$, in terms of soft heaps (*Kaplan et al., 2019*; *Chazelle, 2000*). *Serang (2020)* created an optimal method for selecting the top $k$ values of $A + B$ which does not rely on a pointer-based data structure like the soft heap and is fast in practice. Any reference to "Serang's method" or "Serang's pairwise method" will be in reference to this optimal method which performs pairwise selection on $A + B$.

*Johnson & Mizoguchi (1978)* extended the problem to selecting the $k^{th}$ value in $X_1 + X_2 + \cdots + X_m$ and did so with runtime $O(m \cdot n^{\lceil \frac{m}{2} \rceil} \log(n))$; however, there has not been significant work done on the problem since. *Kreitzberg, Lucke & Serang (2020a)* introduced the FastSoftTree method which performs selection on $X_1 + X_2 + \cdots + X_m$ by creating a balanced binary tree of pairwise selection problems. FastSoftTree uses Kaplan *et al.*'s soft heap-based pairwise selection method and performs selection in $O(n \cdot m + k \cdot m^{\log_2(\alpha^2)})$ time with space usage in $O(n \cdot m + k \log(m))$ for constant $\alpha > 1$. If only the $k^{th}$ value is desired, then Johnson and Mizoguchi's method is the fastest known when $k > \frac{m \cdot n^{\lceil \frac{m}{2} \rceil} \log(n)}{m^{\log_2(\alpha^2)}}$,

Corresponding author
Oliver Serang, oliver.serang@umt.edu

for $k < \frac{m \cdot n^{\lceil \frac{m}{2} \rceil} \log(n)}{m^{\log_2(\alpha^2)}}$ (or if the top $k$ values are desired) Kreitzberg *et al.*'s FastSoftTree is fastest.

Selection on $X_1 + X_2 + \cdots + X_m$ is important for max-convolution (*Bussieck et al., 1994*) and max-product Bayesian inference (*Serang, 2015*; *Pfeuffer & Serang, 2016*). Computing the $k$ best quotes on a supply chain for a business, when there is a prior on the outcome (such as components from different companies not working together), becomes solving the top values of a probabilistic linear Diophantine equation (*Kreitzberg & Serang, 2021*) and thus becomes a selection problem. Finding the most probable isotopologues of a compound such as hemoglobin, $C_{2952}H_{4664}O_{832}N_{812}S_8Fe_4$, may be done by solving $C + H + O + N + S + Fe$, where $C$ would be the most probable isotope combinations of 2,952 carbon molecules (which can be computed via a multinomial at each leaf, ignored here for simplicity), $H$ would be the most probable isotope combinations of 4,664 hydrogen molecules, and so on. The selection method proposed in this paper has already been used to create the world's fastest isotopologue calculator (*Kreitzberg et al., 2020b*).

## Layer-ordered heaps

In a standard binary heap, the only known relationships between a parent and a child is $A_i \leq A_{children(i)}$. A layer-ordered heap (LOH) has stricter ordering than the standard binary heap, but is able to be created in $\Theta\left(n\log(\frac{n}{n \cdot (\alpha - 1) + 1}) + \frac{n \cdot \alpha \cdot \log(\alpha)}{\alpha - 1}\right) = \Theta(n)$ time for constant $\alpha > 1$ (*Pennington et al., 2020*). $\alpha$ is the rank of the LOH and determines how fast the layers grow. A LOH partitions the array into several layers, $L_i$, which grow exponentially such that $|L_1| = 1$ and $\frac{|L_{i+1}|}{|L_i|} \approx \alpha$. Every value in a layer $L_i$ is $\leq$ every value in proceeding layers $L_{i+1}, L_{i+2}, \dots$ which we denote as $L_i \leq L_{i+1}$, this is the "layer-ordering property." If $\alpha = 1$, then all layers are size one and the LOH is sorted; therefore, to be constructed in $\Theta(n)$ time the LOH must have $\alpha > 1$.

## Pairwise selection

Serang's method of selection on $A + B$ utilizes LOHs to be both optimal in theory and fast in practice. In this section we provide a summary of Serang's method, for a more detailed description with analysis see *Serang (2020)*. The method has four phases. Phase 0 is simply to LOHify (make into a layer-ordered heap) the input arrays which can be done in $\Theta(n)$ time.

Phase 1 finds which layer products may be necessary for the selection. A layer product, $A^{(u)} + B^{(v)}$, is the Cartesian product of layers $A^{(u)}$ and $B^{(v)}$: $A_1^{(u)} + B_1^{(v)}, A_2^{(u)} + B_1^{(v)}, \dots, A_1^{(u)} + B_2^{(v)}, \dots$. Finding which layer products are necessary for the selection can be done using a standard binary heap. A layer product is represented in the binary heap in two separate ways: a min tuple $\lfloor u, v \rfloor = (min(A^{(u)} + B^{(v)}), (u, v), false)$ and a max tuple $\lceil (u, v) \rceil = (max(A^{(u)} + B^{(v)}), (u, v), true)$. Creating the tuples does not require calculating the Cartesian product of $A^{(u)} + B^{(v)}$ since $min(A^{(u)} + B^{(v)}) = min(A^{(u)}) + min(B^{(v)})$ which can be found in a linear pass of $A$ and $B$ separately. The same argument applies for $\lceil (u, v) \rceil$. *false* and *true* note that the tuple contains the minimum or maximum value in the layer product, respectively. Also, let *false* = 0 and *true* = 1 so that a min tuple is popped before a max tuple even if they contain the same value.

**(A)** Nine layer products of $A + B$.

|     | B | 4 | 4 10 | 11 17 13 19 |
| --- | --- | --- | --- | --- |
| A   |   |   |   |   |
| 2   | 6 | 6 6 12 | 13 | 21 |
| 2   |   | 6 6 | 13 |   |
| 9   |   | 13 | 19 | 28 |
| 10  |   | 14 14 | 21 |   |
| 15  |   |   |   |   |
| 11  |   |   |   |   |
| 16  |   | 20 | 26 | 35 |

**(B)**

| Val, (u,v), is max? | $\lvert A^u\rvert \cdot \lvert B^v\rvert$ | s |
| --- | --- | --- |
| ( 6,  (1,1), false) | 1 | 0 |
| ( 6,  (1,1), true ) | 1 | 1 |
| ( 6,  (1,2), false) | 2 | 1 |
| ( 6,  (2,1), false) | 2 | 1 |
| ( 6,  (2,2), false) | 4 | 1 |
| (12, (1,2), true ) | 2 | 3 |
| (13, (2,1), true ) | 2 | 5 |
| (13, (1,3), false) | 4 | 5 |
| (13, (2,3), false) | 8 | 5 |
| (14, (3,1), false) | 4 | 5 |
| (14, (3,2), false) | 8 | 5 |
| (19, (2,2), true ) | 4 | 9 |
| (20, (3,1), true ) | 4 | 13 |

**Figure 1** (A): Nine layer products of $A + B$. (B): The layer product tuples in the order they would pop from the heap, the number of values in their Cartesian product, and $s$, the cumulative size of the layer products whose max tuples have been popped. The two axes are the input arrays after being LOHified. The values of all 18 possible layer product tuples are shown (nine min tuples in blue and nine max tuples in green). If $k = 10$, then the tuples will be popped in the order shown in (B). After $(20, (3,1), true)$ is popped, $s$ (the total number of items in the Cartesian product of all max tuples) exceeds $k$. Note that the values in the layers of $A$ and $B$ are not necessarily in sorted order.

Phase 1 uses a binary heap to retrieve the tuples in sorted order. When a min tuple, $\lfloor (u,v) \rfloor$, is popped, then its neighbors ($\lfloor (u+1,v) \rfloor, \lfloor (u,v+1) \rfloor$) and the corresponding max tuple are pushed, assuming $\lfloor (u+1,v) \rfloor, \lfloor (u,v+1) \rfloor$ are in bounds (a set is used to ensure a layer product is inserted only once). When a max tuple is popped, a variable $s$ is increased by $\lvert A^{(u)} + B^{(v)} \rvert = \lvert A^{(u)} \rvert \cdot \lvert B^{(v)} \rvert$ and $(u,v)$ is appended to a list $q$. This continues until $s \geq k$. Figure 1 shows an example of phase 1 when $k = 10$.

In phases 2 and 3 all max tuples still in the heap have their index appended to $q$, then the Cartesian product of all layer products in $q$ are generated. A linear time one-dimensional $k$-select is performed on the values in the Cartesian products to produce only the top $k$ values in $A + B$. The time complexity of the algorithm is linear in the overall number of values produced which is $O(k)$.

In this paper we efficiently perform selection on $X_1 + X_2 + \cdots + X_m$ by utilizing Serang's pairwise selection method.

## METHODS

In order to retrieve the top $k$ values from $X_1 + X_2 + \cdots + X_m$, a balanced binary tree of pairwise selections is constructed. The top $k$ values are calculated by selection on $X_1 + X_2, X_3 + X_4, \ldots$ then on $(X_1 + X_2) + (X_3 + X_4), (X_5 + X_6) + (X_7 + X_8), \ldots$. All data loaded and generated is stored in arrays which are contiguous in memory, allowing for great cache performance compared to a soft heap-based method.

## Tree construction

The tree has height $\lceil \log_2(m) \rceil$ with $m$ leaves, each one is a wrapper around one of the input arrays which are unsorted and have no restrictions on the type of data they contain. Upon construction, the input arrays are LOHified in $\Theta(n \cdot m)$ time, which is amortized out by the cost of loading the data. Each node in the tree above the leaves performs pairwise selection on two LOHs: one generated by its left child and one generated by its right child. All nodes in the tree generate their own LOH, but this is done differently for the leaves vs the pairwise selection nodes. When a leaf generates a new layer it simply allows its parent to have access to the values in the next layer of the LOHified input array. For a pairwise selection node, generating a new layer is more involved.

## Pairwise selection nodes

Each node above the leaves is a pairwise selection node, each of which has two children that may be leaves or other pairwise selection nodes. In contrast to the leaves, the pairwise selection nodes will have to calculate all values in their LOHs by generating an entire layer at a time. Generating a new layer requires performing selection on $A + B$, where $A$ is the LOH of its left child and $B$ is the LOH of its right child. Due to the combinatorial nature of this problem, simply asking a child to generate their entire LOH can be exponential in the worst case so they must be generated one layer a time and only as necessary.

The pairwise selection performed is Serang's method with a few modifications (though Fig. 1 is still representative of the method). The size of the selection is always the size of the next layer, $k = |L_i|$, to be generated by the parent. The selection begins in the same way as Serang's: a heap is used to pop min and max layer product tuples. When a min tuple, $\lfloor (u, v) \rfloor$, is popped the values in the Cartesian product are generated and appended to a list of values to be considered in the $k$-selection. The neighboring layer products inserted into the heap are determined using the scheme from Kaplan *et al.* which differs from Serang's method: $\lceil (u, v) \rceil$ and $\lfloor (u, v + 1) \rfloor$ are always inserted and, if $v = 1$, $\lfloor (u + 1, v) \rfloor$ is inserted as well. This insertion scheme will not repeat any indices and therefore does not require the use of a set to keep track of the indices in the heap. When any min tuple is proposed, the parent asks both children to generate the layer if it is not already available. If one or both children are not able to generate the layer (i.e., the index is larger than the full Cartesian product of the child's children) then the parent does not insert the tuple into its heap. The newly generated layer is simply appended to the parent's LOH and may now be accessed by the parent's parent. An example of a pairwise selection node generating a new layer which requires a child to generate a new layer is shown in Fig. 2.

The dynamically generated layers should be kept in individual arrays, then a list of pointers to the arrays may be stored. This avoids resizing a single array every time a new layer is generated.

Theorem 1 in *Serang (2020)* proves that Serang's method performs selection in $O(k)$ time. lemmas 6 and 7 show that the number of all values generated is $O(n + k)$; however, lemma 7 may be amended to show that any layer product of the form $(u, 1)$ or $(1, v)$ will generate $\leq \alpha \cdot |(u - 1, 1)| \in O(k)$ or $\leq \alpha \cdot |(1, v - 1)| \in O(k)$ values, respectively, to show

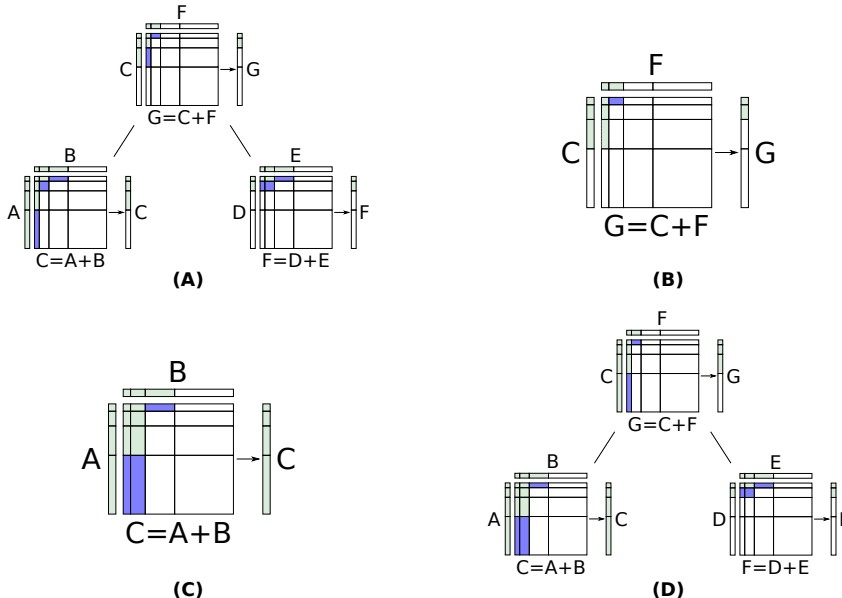

**Figure 2** **The process of adding a layer to a pairwise selection node's LOH, *G*, when both children are pairwise selection nodes. Each node has a LOH it generates for its parent to access as well as a (not realized) matrix formed by the Cartesian product of its children's LOHs.** Blue layer products currently have their min tuple in the heap. Green layer products have had at least their min tuple popped from the heap (and thus have inserted other tuples into the heap). (A) The triplet before adding a new layer to *G*. (B) The parent generating the next layer in its LOH. The parent pops $\lfloor(1,3)\rfloor$ and must now insert $\lceil(1,3)\rceil$ and $\lfloor(1,4)\rfloor$; however, the left child has not yet generated the fourth layer in its LOH, *C*, so the parent can not insert $\lfloor(1,4)\rfloor$. (C) The left child generating the fourth layer in its LOH, *C*. The left child pops $\lfloor(2,2)\rfloor$ then inserts $\lceil(2,2)\rceil$ and $\lfloor(2,3)\rfloor$. The left child continues and pops $\lceil(2,2)\rceil$ and $\lfloor(2,3)\rfloor$ and performs the appropriate insertions: $\lceil(2,3)\rceil$ and $\lfloor(2,4)\rfloor$. Finally, the left child pops $\lceil(2,3)\rceil$ at which point it has enough values to select the next layer in *C*. Now that *C* has its fourth layer, the parent is able insert $\lfloor(1,4)\rfloor$ and continue. The parent then pops $\lceil(1,3)\rceil$ and selects its third layer. The parent did not need *F* to generate a new layer and so the right child remains the same as before. (D) The triplet after the parent and left child perform the necessary operations to generate the next layer in *G*.

that the total values generated is $O(k)$. Thus, the total number of values generated when a parent adds a new layer $L_i$ is $O(|L_i|)$.

## Selection from the root

In order to select the top $k$ values from $X_1 + X_2 + \cdots + X_m$, the root is continuously asked to generate new layers until the cumulative size of the layers in their LOH exceeds $k$. Then a $k$-selection is performed on the layers to retrieve only the top $k$. Due to the layer-ordering property, a selection only needs to be performed on the last layer, all values in previous layers will be in the top $k$.

The Cartesian product tree is constructed in the same way as the FastSoftTree (*Kreitzberg, Lucke & Serang, 2020a*) as both methods dynamically generate new layers in a similar manner with the same theoretical runtime. In both methods, the pairwise selection creates at most $O(\alpha^2 \cdot k)$ values. Thus the theoretical runtime of both methods is $O(n \cdot m + k \cdot m^{\log_2(\alpha^2)})$ with space usage $O(n \cdot m + k \log(m))$.

### Wobbly version

In Serang's pairwise selection, after enough layer product tuples are popped from the heap to ensure they contain the top $k$ values, there is normally a selection performed to remove any excess values. Strictly speaking, this selection is not necessary anywhere in the tree except for the root when the final $k$ values are returned. When the last max tuple $\lceil(u, v)\rceil$ is popped from the heap, $max(A^{(u)} + B^{(v)})$ is an upper bound on the $k^{th}$ value in the $k$-selection. Instead of doing a $k$-selection and returning the new layer, which requires a linear time selection followed by a linear time partition, we can simply do a value partition on $max(A^{(u)} + B^{(v)})$.

A new layer generated from only a value partition and not a selection is not guaranteed to be size $k$, it is at least size $k$ but contains all values $\leq max(A^{(u)} + B^{(v)})$. In the worst case, this may cause layer sizes to grow irregularly with a constant larger than $\alpha$. For example, if $k = 2$ and $|L_1| = |L_2| = 1$ then in the worst case every parent will ask their children to each generate two layers and the value partition will not remove any values. Each leaf will generate two values, their parents will then have a new layer of size $2^2 = 4$, their parent will have a new layer of size $(2^2)^2 = 16$, etc. Thus the root will have to perform a 2-selection on $2^m$ values which will be quite costly.

In an application like calculating the most probable isotopologues of a compound, this version can be quite beneficial. For example, to generate a significant amount of the isotopologues of the titin protein may require $k$ to be hundreds of millions. Titin is made of only carbon, hydrogen, nitrogen, oxygen, and sulfur so it will only have five leaves and a tree height of three. The super-exponential growth of the layers for a tree with height three is now preferential because it will still not create so many more than $k$ values but it will do so in many fewer layers with only value partitions and not the more costly linear selections. We call this the "wobbly" Cartesian product tree.

## RESULTS

All experiments were run on a workstation equipped with 256GB of RAM and two AMD Epyc 7351 processors running Ubuntu 18.04.4 LTS. The data was randomly generated using the built-in `rand()` function (seeded with the value 1) in C++. All values were integers between 0 and 10,000. Though arrays of random values are used, the performance gain does not depend on the values in the arrays. If the input arrays are homogeneous so that the value for all min and max tuples are the same, they will be popped in ascending order of their index tuple and will still generate at most $O(\alpha^2 \cdot k)$ values.

### Cartesian product trees vs FastSoftTree

In a Cartesian product tree, replacing the pairwise $A + B$ selection steps from Kaplan *et al.*'s soft heap-based algorithm with Serang's optimal LOH-based method provides the same $o(n \cdot m + k \cdot m)$ theoretical performance for the Cartesian product tree but is practically much faster (Table 1). This is particularly true when $k \cdot m^{\log_2(\alpha^2)} \gg n \cdot m$, where popping values dominates the cost of loading the data. When $k \geq 2^{10}$, $k \cdot m^{0.2750} < n \cdot m$ which is reflected in our results where for $k = 2^{20}$ we get a 630.4× speedup, significantly larger than for $k = 2^{10}$ which only has a 10.27× speedup. For any reasonable $\epsilon$ and $\alpha$,

**Table 1  Runtimes for Cartesian product tree vs FastSoftTree with $n = 32$, $m = 256$ and $\alpha = 1.1$.** The runtime is averaged over 20 iterations. For small problems the soft heap-based tree is competitive with the Cartesian product tree; however, for large enough k the cache performance of the LOH significantly outperforms the soft heap resulting in a 630.4 × speedup for $k = 2^{20}$.

| k | Cartesian product tree (seconds) | FastSoftTree (seconds) |
|---|---|---|
| $2^2$ | $1.404 \times 10^{-03}$ | $3.146 \times 10^{-03}$ |
| $2^3$ | $1.504 \times 10^{-03}$ | $2.855 \times 10^{-03}$ |
| $2^4$ | $1.521 \times 10^{-03}$ | $3.163 \times 10^{-03}$ |
| $2^5$ | $1.592 \times 10^{-03}$ | $2.618 \times 10^{-03}$ |
| $2^6$ | $1.689 \times 10^{-03}$ | $4.172 \times 10^{-03}$ |
| $2^7$ | $1.718 \times 10^{-03}$ | $4.830 \times 10^{-03}$ |
| $2^8$ | $1.881 \times 10^{-03}$ | $8.864 \times 10^{-03}$ |
| $2^9$ | $2.080 \times 10^{-03}$ | 0.01143 |
| $2^{10}$ | $1.745 \times 10^{-03}$ | 0.01792 |
| $2^{11}$ | $2.217 \times 10^{-03}$ | 0.02362 |
| $2^{12}$ | $3.123 \times 10^{-03}$ | 0.04459 |
| $2^{13}$ | $3.318 \times 10^{-03}$ | 0.07026 |
| $2^{14}$ | $5.099 \times 10^{-03}$ | 0.111 |
| $2^{15}$ | $6.240 \times 10^{-03}$ | 0.2296 |
| $2^{16}$ | $8.724 \times 10^{-03}$ | 0.4952 |
| $2^{17}$ | 0.01266 | 0.9609 |
| $2^{18}$ | 0.01663 | 1.610 |
| $2^{19}$ | 0.02684 | 12.77 |
| $2^{20}$ | 0.0405 | 25.53 |

the difference in number of values generated (and thus in memory usage) will not differ significantly. Regardless of which algorithm generates less values, the performance gain due to generating less overall values will be negligible compared to the difference in cache performance between soft heaps and LOHs.

## Standard vs Wobbly

As we see in Table 2, for small $m$ the Cartesian product tree can gain significant increases in performance when there are no linear selections performed in the tree and the layers are allowed to grow super-exponentially. As $k$ grows, the speedup of the wobbly version continues to grow, resulting in a 1.786× speedup for $k = 2^{30}$. When $m \gg 5$ the growth of the layers near the root start to significantly hurt the performance. For example, if $n = 32, m = 256$ and $k = 256$ the wobbly version takes 0.5805 seconds and produces 149,272 values at the root compared to the non-wobbly version which takes $1.8810 \times 10^{-03}$ seconds and produces just 272 values at the root.

## DISCUSSION

Replacing pairwise selection which uses a soft heap with Serang's method provides a significant increase in performance. Since both methods LOHify the input arrays (using the same LOHify method) the most significant increases are seen when $k \cdot m^{\log_2(\alpha^2)} \gg n \cdot m$.

**Table 2   Runtimes for standard Cartesian product tree vs wobbly Cartesian product tree with $n = 256$, $m = 5$ and $\alpha = 1.1$.** The runtime is averaged over 20 iterations for the two methods. With $m = 5$ the tree only has three layers and so the super-exponential growth of the layers as they go from the leaves to the roots does not become intractable. As $k$ becomes extremely large the ability of the wobbly tree to generate huge layers at the root without performing any selections significantly reduces the runtime resulting in a $1.786 \times$ speedup.

| k | Standard version (seconds) | Wobbly version (seconds) |
|---|---|---|
| $2^2$ | $1.544 \times 10^{-04}$ | $1.777 \times 10^{-04}$ |
| $2^3$ | $1.754 \times 10^{-04}$ | $1.468 \times 10^{-04}$ |
| $2^4$ | $2.086 \times 10^{-04}$ | $1.846 \times 10^{-04}$ |
| $2^5$ | $2.386 \times 10^{-04}$ | $2.046 \times 10^{-04}$ |
| $2^6$ | $2.080 \times 10^{-04}$ | $1.935 \times 10^{-04}$ |
| $2^7$ | $3.060 \times 10^{-04}$ | $2.672 \times 10^{-04}$ |
| $2^8$ | $3.481 \times 10^{-04}$ | $3.225 \times 10^{-04}$ |
| $2^9$ | $4.289 \times 10^{-04}$ | $2.978 \times 10^{-04}$ |
| $2^{10}$ | $6.119 \times 10^{-04}$ | $4.087 \times 10^{-04}$ |
| $2^{11}$ | $7.976 \times 10^{-04}$ | $4.585 \times 10^{-04}$ |
| $2^{12}$ | $1.000 \times 10^{-03}$ | $7.263 \times 10^{-04}$ |
| $2^{13}$ | $1.711 \times 10^{-03}$ | $1.189 \times 10^{-03}$ |
| $2^{14}$ | $2.344 \times 10^{-03}$ | $1.465 \times 10^{-03}$ |
| $2^{16}$ | $7.531 \times 10^{-03}$ | $4.890 \times 10^{-03}$ |
| $2^{15}$ | $3.919 \times 10^{-03}$ | $2.578 \times 10^{-03}$ |
| $2^{17}$ | 0.0113 | $9.090 \times 10^{-03}$ |
| $2^{18}$ | 0.01741 | 0.01583 |
| $2^{19}$ | 0.02777 | 0.02511 |
| $2^{20}$ | 0.04904 | 0.04228 |
| $2^{21}$ | 0.08572 | 0.07773 |
| $2^{22}$ | 0.1623 | 0.1424 |
| $2^{23}$ | 0.3274 | 0.234 |
| $2^{24}$ | 0.636 | 0.4838 |
| $2^{25}$ | 1.210 | 1.029 |
| $2^{26}$ | 2.306 | 1.588 |
| $2^{27}$ | 4.993 | 3.487 |
| $2^{28}$ | 9.995 | 8.441 |
| $2^{29}$ | 19.7 | 14.31 |
| $2^{30}$ | 43.45 | 24.33 |

For small $m$, the performance can be boosted using the wobbly version; however, for large $m$ the super-exponentially sized layers can quickly begin to dampen performance. It may be possible to limit the layer sizes in the wobbly version by performing selections only at certain layers of the tree: either by performing the selection on every $i^{th}$ layer or only on the top several layers.

For all experiments $\alpha = 1.1$, though the optimal $\alpha$ is not known. As $\alpha$ approaches 1 the layer sizes also approach 1 and the LOHs become fully sorted. Having all layer sizes of 1 will mean not producing any unnecessary values; however, this will lead to slower practical

performance due to popping $k$ layer product tuples from the binary heap which will cause runtime to be in $\Omega(k\log(k))$. As $\alpha$ grows, both the number of pops from the binary heap and the time to LOHify the input arrays decreases but the amount of unnecessary values increases. The unnecessary values will slow practical performance both when they are calculated and in the subsequent linear selections and partitions performed to create the layer. In practice, $\alpha = 1.1$ seems to find a good balance between number of pops from binary heaps and the number of unnecessary values generated, both in Tables 1 and 2 as well as when calculating the most probable isotopologues of a compound.

Due to corruption, when using a soft heap-based method, online computation of values requires retrieving the top $k_1$ values and then top $k_2$ remaining values and so on in order to be performed with optimal runtime. Selection on $X_1 + X_2 + \cdots + X_m$ requires optimal, online computation of values in each pairwise selection node in order to be performed in $o(n \cdot m + k \cdot m)$ time, but this applies to any soft heap-based method which requires online computation of the top values.

The number of corrupt values is bounded by $\epsilon \cdot I$, where $I$ insertions have been performed to date; therefore, there are at most $\epsilon \cdot c \cdot k_1$ corrupt values. The top $k_1$ values can be selected by retrieving no more than $k_1 + \epsilon \cdot c \cdot k_1$ values from the soft heap and then performing a $k_1$-selection (via median-of-medians) on the retrieved values. The $\epsilon \cdot k_1$ corrupt values are reinserted into the soft heap, bringing the total insertions to $k_1 \cdot \epsilon \cdot (1 + c)$. To retrieve the top $k_2$ remaining values, $k_2 + \epsilon \cdot k_1 \cdot \epsilon \cdot (1 + c) \in \Omega(k_2 + k_1)$ values need to be popped. These top $k_2$ values can be retrieved in optimal $O(k_2)$ time if $k_2 \in \Theta(k_1)$. Likewise, $k_3 \in \Theta(k_1 + k_2)$, and so on. Thus, for optimal, online computation the sequence of $k_j$ values must grow exponentially.

Rebuilding the soft heap (rather than reinserting the corrupted values into the soft heap) instead does not alleviate this need for exponential growth in $k_1, k_2, \ldots$ required to achieve optimal $O(k_1 + k_2 + \cdots)$ total runtime. When rebuilding, each next $k_j$ must be comparable to the size of the entire soft heap (so that the cost of rebuilding can be amortized out by the optimal $\Theta(k_j)$ steps used to retrieve the next $k_j$ values). Because $c \geq 1$, the size of the soft heap is always $\geq k_1 + k_2 + \cdots + k_{j-1}$ for the selections already performed, and thus the rebuilding cost is $k_1 + k_2 + \cdots + k_{j-1}$, which must be $\in \Theta(k_j)$. This likewise requires exponential growth in the $k_j$.

The necessity to have exponential growth in $k_j$ enforces the layer-ordering property on the resulting values. It is the layer-ordering property which guarantees that a proposal scheme such as that in Kaplan *et al.* does not penetrate to great depth in the combinatorial heap, which could lead to exponential time complexity when $c > 1$. In this manner, the $k_1, k_2, \ldots$ values can be seen to form layers of a LOH, which would not require retrieving further layers before the current extreme layer has been exhausted. Thus, both FastSoftTree and Cartesian product trees have to have pairwise selection nodes which can generate LOHs one layer at a time. FastSoftTree nodes require a soft heap to generate the LOHs where Cartesian product tree nodes generate layers in a straight-forward manner.

This method has already proved to be beneficial in generating the top $k$ isotopologues of chemical compounds, but it is not limited to this use-case. It is applicable to fast algorithms for inference on random variables $Y = X_1 + X_2 + \cdots + X_m$ in the context of graphical

Bayesian models. It may not generate a value at every index in a max-convolution, but it may generate enough values fast enough to give a significant result.

**Code availability**

A C++ implementation can be found in https://bitbucket.org/seranglab/cartesian-product-tree/ under the MIT license. The code is free for academic and commercial use.

### Funding

This work was supported by NSF CAREER grant 1845465. The funders had no role in study design, data collection and analysis, decision to publish, or preparation of the manuscript.

### Grant Disclosures

The following grant information was disclosed by the authors:
NSF CAREER: 1845465.

### Competing Interests

The authors declare there are no competing interests.

### Author Contributions

- Patrick Kreitzberg, Kyle Lucke, Jake Pennington and Oliver Serang conceived and designed the experiments, performed the experiments, analyzed the data, performed the computation work, prepared figures and/or tables, authored or reviewed drafts of the paper, and approved the final draft.

### Data Availability

Software implementations of the code are available at BitBucket: https://bitbucket.org/seranglab/cartesian-product-tree.

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
