# Peer review of "Selection on X1 + X2 + ⋯ + Xm via Cartesian product trees"

_PeerJ Computer Science, doi:10.7717/peerj-cs.483_

## Round 0.1 · original submission · Minor Revisions

Please update the manuscript according to the reviews. Please address the specific points listed.

·

Basic reporting

The authors describe rather complex algorithms and data structures in an easy but professional language. English grammar and spelling are (to my knowledge) flawless. The authors improve a well-defined, very elemental problem, for which the amount of literature references is sufficient, also because a short recap is provided in the introduction.

However, since there are a few variations of the algorithm (and its "parts") described, it sometimes was not immediately clear to me which exact variation was referred to in later parts of the manuscript.
E.g. in line 70 "Serang's pairwise selection method" is mentioned. I am confident that the method from Serang (2020) is meant here but I think it would be beneficial if these methods would be introduced (e.g. in line 25) with a certain consistent name or phrase that is referred to later.
Similar to "FastSoftTree" (L. 117), for which I did not find any occurrence in the introduction, though.
In general, maybe a little Table in the Methods would help to clarify which combination of the referenced (or newly developed) algorithms/data structures (for storing the tuples, for making new proposals, for building the tree, for making wobbly selections [if applicable]) are used in "FastSoftTree" vs the new "Cartesian product tree".

In line 63 the authors speak of a "neighboring layer". Until I saw the figure, I was wondering if a "diagonally touching" (i.e. u+1, v+1) layer is neighboring. This could be clarified from the beginning.

I really liked figure 1, it helps a lot to understand the concepts. I would, however, suggest adding a sentence to its caption on why "the total number of items (...) is >= 10". Alternatively, another column for the size of the Cartesian product represented by the tuples or a short reminder that the size of each is u*v. This would make it more self-contained.

Since figure 1 helped so much, I was wondering if the authors could add a similar figure for the actual extension to n (e.g. n=3) vectors, that is described in the manuscript. I am myself not sure if it can be broken down into a simple figure or if this is asking too much, so this is a mere suggestion.

Experimental design

The research question itself is relatively simple and well defined here but it is of great importance to a lot of methods. The extensions/improvements to existing algorithms provided in the article is a welcome addition to the landscape of methods and described in detail. The evaluation of the presented algorithm is sufficient. The speed improvement in practice is clear.

It seems like it was not mentioned in the text, how the test data was exactly generated. On bitbucket "srand" is mentioned. It would be good if this would be shortly clarified in the text. The used random seed could be mentioned but results will probably differ on other platforms anyway.

I was also missing a short comment about memory consumption. They are the same in theory but are they (roughly) the same in practice? It does not necessarily need to be evaluated if it is similar/negligible.

Lastly, a small suggestion: Does the number of data points in the Tables already justify a simple line plot? I think it would significantly reduce the time needed for readers to see the large improvements.

Validity of the findings

The source code is provided and I managed to compile and run it. I am confident that the results can be reproduced (through e.g. the seed for random numbers).

Regarding the Discussion of the results, I was struggling a bit to put LL167-189 of the Discussion into context. Which exact problem, observation, or use case does it discuss?

I was also wondering if the composition of the vectors has any impact on the algorithms. Specifically, do long stretches of equal numbers have any impact? Maybe this can shortly be discussed if I am not missing something obvious.

I assume that the speed-up translates to different alphas? It would be nice if the authors could at least provide an argument on why this should be the case, otherwise, it would probably be good to include another alpha to the test data.

Additional comments

Minor potential errors:
- L 115 "cumulative size the layers": There is probably an "of" missing.
- L 118 unclear usage of "both"
- L 134 "(2^2)^2 = 8": Should be 16 (actually depends on the level of the second child I guess but from context, a full tree is assumed here)
- L 148 "(Table 1": Closing parentheses missing
- L 149 and 162 "m^log2(alpha^2)": Is this supposed to be log_2 in subscript? As in L 120?

Reviewer 2 ·

Basic reporting

I found the basic reporting of this paper to be largely professional and clear. However, I have collected a list of points in the paper that were unclear or needed more references.

Issues of clarity and correctness:
Lines 20, 21, 45, 49, 68, 108, 109: I suggest specifying what computational resource the asymptotic complexity notation you use is referring to. (Examples: in Line 20, replace “takes O(n^2 \log(n))” with “takes O(n^2 \log(n)) time” and in line 21, replace “no O(n^2) algorithm” with “no O(n^2) time algorithm”.)
Line 44-45: The phrase “is able to be created in” suggests an algorithm (i.e. upper bound), but instead you use Omega, suggesting a lower bound. Perhaps the LOH can be created in O(n) time and this is optimal? Please clarify.
Line 80: Since there are m input arrays, wouldn’t it take O(mn) time to LOHify all the input arrays? Please clarify.
Line 109: Capitalize “lemma”
Line 115: replace “the cumulative size the layers” with “the cumulative size of the layers”
Line 148: Right parentheses needed after “(Table 1”
Line 177: The authors state that “the sequence of k values must grow exponentially”. I believe they mean that in order for the online retrieval of top-k values to take the optimal amount of time, the values k_1, k_2, … must grow exponentially. Please clarify whether this is the case. As I understand it, the k_j values are given to the algorithm at runtime, so they don’t necessarily have to grow exponentially, as stated. But they do need to grow exponentially for the algorithms given in the paragraphs at 170 and 178 to be optimal. I believe this is an important distinction that should be made.


Issues of literature references and context:
Line 29-30: What algorithm is faster for small k? Is there a trivial, faster solution for small k? This may be worth clarifying in any revisions.
Line 50: Here the authors describe “Serang’s method of selection on A+B”. If I understand correctly, “Serang’s method” refers to the algorithm presented in Serang (2020). Please clarify whether this is the method being summarized in this section with a citation. It would be useful to readers to be able to know which text to reference in order to better understand Serang’s method.
Line 63: What are the “neighboring layer products” mentioned here? I understand that the procedure described in this section is presented fully in another paper. However, I believe the authors should either give a more complete summary of Serang’s method of selection on A+B or else state explicitly that they are summarizing and make a note of where full details can be found.

Experimental design

I have found the experimental design of this paper largely satisfactory with the exception of a few specific questions about the experimental method and results that should be clarified in revisions.

1. How were the sets X_1, X_2, …, X_m initialized for the experimental evaluations given in Table 1 and Table 2? This information should be given for replication purposes and because it could impact the performance of algorithm implementations.

2. What are the units used in Table 1 and Table 2 for experimental runtime? I assume seconds, but please state it somewhere.

Validity of the findings

The authors found that their algorithm implementations were faster at selection than FastSoftTree, and in particular that Wobbly version of their Cartesian product tree implementation yielded significant speedups over their standard algorithm when m is small and k is large. They also give a detailed comparison of possible approaches to the problem of online computation of top k values in a soft heap and describe why these approaches fail to produce optimal algorithms. However, I find the argument given in the paragraph at 185 about the “layer ordering property” to be vague and too ambiguous to evaluate scientifically. Please consider clarifying your description of the "layer ordering property" and how it pertains to selection on X_1 + ... + X_m. If the layer ordering property is a more speculative aspect of the Discussion, please identify this in your paper.

---

## Round 0.2 · accepted · Accept

Please take care of the minor issue noted by the reviewer before publication.

·

Basic reporting

I agree that the more detailed introduction/separation of the referenced algorithms suffices. A table should not be necessary.
Thank you a lot for the additional figure, it is everything that I hoped for. I also agree that there is no need to cover more complex scenarios.

Experimental design

The additional clarifications that were added helped a lot and addressed all my concerns. It should also be easier to replicate now.
I understand the concerns about transforming the table into a line plot and see the benefits of the current state.

Validity of the findings

The discussion was significantly improved. It is easier to follow and discusses the few points that were in my opinion missing before.

Additional comments

Potential minuscule errors, if they can be addressed before potential publication:
L31: "et al." not in italics like in the rest of the block (just for consistency)
L36: "Kreitberg" instead of "Kreitzberg"?